# Clinical manifestations of Rift Valley fever in humans: Systematic review and meta-analysis

Zacchaeus Anywaine[1,2]*, Swaib Abubaker Lule[3], Christian Hansen[2,4], George Warimwe[5,6], Alison Elliott[1,2]

1 Department of Clinical Research, London School of Hygiene and Tropical Medicine, London, United Kingdom, 2 Medical Research Council/Uganda Virus Research Institute and London School of Hygiene and Tropical Medicine Uganda Research Unit, Entebbe, Uganda, 3 Institute for Global Health, University College London, London, United Kingdom, 4 MRC International Statistics & Epidemiology Group, London School of Hygiene and Tropical Medicine, London, United Kingdom, 5 Centre for Tropical Medicine and Global Health, University of Oxford, Oxford, United Kingdom, 6 KEMRI WellcomeTrust Research Programme, Kilifi, Kenya

* Zacchaeus.Anywaine@mrcuganda.org

## Abstract

### Background

Rift Valley fever (RVF) is an emerging, neglected, mosquito-borne viral zoonosis associated with significant morbidity, mortality and expanding geographical scope. The clinical signs and symptoms in humans are non-specific and case definitions vary. We reviewed and analysed the clinical manifestations of RVF in humans.

### Methods

In this systematic review and meta-analysis we searched on different dates, the Embase (from 1947 to 13th October 2019), Medline (1946 to 14th October 2019), Global Health (1910 to 15th October 2019), and Web of Science (1970 to 15th October 2019) databases. Studies published in English, reporting frequency of symptoms in humans, and laboratory confirmed RVF were included. Animal studies, studies among asymptomatic volunteers, and single case reports for which a proportion could not be estimated, were excluded. Quality assessment was done using a modified Hoy and Brooks et al tool, data was extracted, and pooled frequency estimates calculated using random effects meta-analysis.

### Results

Of the 3765 articles retrieved, less than 1% (32 articles) were included in the systematic review and meta-analysis. Nine RVF clinical syndromes were reported including the general febrile, renal, gastrointestinal, hepatic, haemorrhagic, visual, neurological, cardio-pulmonary, and obstetric syndromes. The most common clinical manifestations included fever (81%; 95% Confidence Interval (CI) 69–91; [26 studies, 1286 patients]), renal failure (41%; 23–59; [4, 327]), nausea (38%; 12–67; [6, 325]), jaundice (26%; 16–36; [15, 393]), haemorrhagic disease (26%; 17–36; [16, 277]), partial blindness (24%; 7–45; [11, 225]), encephalitis (21%; 11–33; [4, 327]), cough (4%; 0–17; [4, 11]), and miscarriage (54%) respectively.

**Data Availability Statement:** All data extraction files are available from the London School of Hygiene and Tropical Medicine (LSHTM) database (DOI: doi.org/10.17037/DATA.00002178).

**Funding:** This work was conducted at the MRC/UVRI and LSHTM Uganda Research Unit which is jointly funded by the UK Medical Research Council (MRC) and the UK Department for International Development (DFID) under the MRC/DFID Concordant agreement and is also part of the EDCTP2 programme supported by the European Union. This project was funded by the UK Biotechnology and Biological Sciences Research Council (BBSRC) and the Medical Research Council (MRC)/Department of Health, through the UK Vaccines Network which is a Government Funding Stream (Grant no: 16/107/02). SL was supported by the PANDORA-ID-NET Consortium (EDCTP Reg/Grant RIA2016E-1609) funded by the European and Developing Countries Clinical Trials Partnership (EDCTP2) programme under the Horizon 2020, the European Union's Framework Programme for Research and Innovation. GMW is supported by an Oak foundation fellowship and a Wellcome Trust grant (grant number 203077_Z_16_Z). The funders had no role in study design, data collection and analysis, decision to publish, or preparation of the manuscript.

**Competing interests:** The authors have declared that no competing interests exist.

Death occurred in 21% (95% CI 14–29; [16 studies, 328 patients]) of cases, most of whom were hospitalised.

## Discussion

This study delineates the complex symptomatology of human RVF disease into syndromes. This approach is likely to improve case definitions and detection rates, impact outbreak control, increase public awareness about RVF, and subsequently inform 'one-health' policies. This study provides a pooled estimate of the proportion of RVF clinical manifestations alongside a narrative description of clinical syndromes. However, most studies reviewed were case series with small sample sizes and enrolled mostly in-patients and out-patients, and captured symptoms either sparsely or using broad category terms.

## Author summary

Rift Valley fever (RVF) is a neglected, arboviral zoonosis that causes severe and diverse disease manifestations in humans. Currently no licenced vaccines exist for use in humans and there is limited surveillance and estimation of disease burden which in part is due to the inability to concisely define the disease. We searched Medline, Embase, Global Health, and Web of Science for published reports on the clinical manifestations of RVF in humans. Studies published in English, reporting frequency of symptoms in humans, and laboratory confirmed RVF were included. We excluded animal studies, studies among asymptomatic volunteers and single case reports for which a proportion could not be estimated. Pooled symptom frequency estimates were calculated using random effects meta-analysis. This review provides a detailed aggregation of the relative frequency of symptoms, and a description of the RVF clinical manifestations in humans. Previous systematic reviews provided a narrative account and it was difficult to identify the most relevant features of RVF disease in areas where other endemic infections present with similar symptoms. This review will refine the clinical diagnosis, improve case detection, and increase public awareness about RVF presentation in humans.

## Introduction

Rift Valley fever (RVF) is a neglected, mosquito-borne and direct contact viral zoonosis associated with significant morbidity, mortality and an expanding geographical scope [1]. The first documented outbreak occurred in Kenya in 1930 [2], but by the turn of the millennium the virus was widely recognised in African countries [3,4]. In the year 2000, a major outbreak occurred for the first time in the Arabian Peninsula [5,6], and in 2016 one case of RVF initially thought to be yellow fever was imported into China [7]. Human cases of accidental laboratory infections with RVF have been reported in non-endemic countries such as the United States of America (USA) [8–10] and United Kingdom (UK) [11]. Recent studies have shown the virus has potential for global epidemics [12,13] and use in bioterrorism [14].

The symptoms of RVF are non-specific since they are consistent with many endemic tropical infections [15]. Further, majority of laboratories in areas prone to RVF outbreaks have inadequate diagnostic capacity [5,16,17], contributing to an underestimation of the RVF disease burden [18].

In Africa, human RVF surveillance is low [4,19], and disease case definitions have great variability [5,20–23]. There is currently no detailed description of the relative frequency of clinical manifestations of RVF disease in humans. Having such a description would guide clinicians on the most common disease features to look out for in patients during RVF outbreaks and routine surveillance. We conducted a systematic review and meta-analysis of existing literature aimed at determining the frequency and scope of clinical and laboratory manifestations of RVF in humans.

## Methods

### Searching strategy

This systematic review was conducted following a protocol registered on the Prospero international prospective register of systematic reviews (PROSPERO) at https://www.crd.york.ac.uk/prospero/, (ID: CRD42019128928). Four electronic databases including Embase (1947 to 13[th] October 2019), Medline (1946 to 14[th] October 2019), Global Health (1910 to 15[th] October 2019) and Web of Science (1970 to 15[th] October 2019) were searched for publications. The search was performed on a combination of key concepts including Rift Valley fever, clinical manifestations and Africa/Arabian Peninsula. Medical and non-medical search terms and synonyms, truncations, wildcards, proximity operators, free text and medical subject headings (MESH) were used in the search. Boolean operators "OR" and "AND" were used to link search terms within and between search concepts respectively. The review protocol and amendments, main search concepts and respective search terms used are available on the PROSPERO database at https://www.crd.york.ac.uk/PROSPEROFILES/128928_STRATEGY_20191023.pdf. The detailed search strategy in each database are provided in S1–S4 Tables.

### Study selection

Search results from each database were exported to Endnote (Thomson Reuters, version X7), duplicates removed and the remaining articles further exported into Microsoft Excel (Microsoft Corporation). Two independent reviewers (ZA and SL) assessed titles and abstracts for full-text review and abstracted data for synthesis. Inconsistencies were discussed and consensus reached at each stage of the selection process (selection, quality assessment and data extraction). Reference lists of all eligible articles were screened to identify additional eligible papers. The outputs were summarised in a flow diagram. Eligible articles included studies published in English that reported on the frequency of RVF symptoms and laboratory abnormalities in humans with laboratory confirmed RVF. There was no restriction on the age of participants. Animal studies, studies among asymptomatic volunteers and single case reports for which a proportion could not be estimated were excluded.

### Data extraction

Data were extracted independently by two authors (ZA and SL) using a standardised Microsoft Excel data extraction spreadsheet. The relative frequency of signs, symptoms and laboratory abnormalities were recorded. For case series, a tally of symptoms in different patients was made and frequencies calculated. Qualitative information on the characteristics of each symptom was extracted on a separate Microsoft Excel spreadsheet to maintain the uniformity in recording. Data was captured on the first author's surname, year of publication, countries where study was conducted, study design, demographics (age, and sex), and source of patients (in-patients, out-patients, in-patients and out-patients, or community patients). For "In-patients", the subjects source in the study were solely hospital based patients requiring

admission; "Out-patients", the subjects source in the study were only hospital based patients requiring no admission; "In-patients and out-patients", the subjects source in the study were both hospital based patients requiring admission and patients requiring no admission; whereas "Community patients" referred to subjects whose source in the study was non-hospital based patients found in the community or at home. The rationale for this stratification by the reviewers was to depict the level of disease severity among patients included in the studies. In addition, data were captured on variables such as laboratory tests used to confirm the diagnosis of RVF, number or proportion that had co-infections and the type of co-infections. Where more than one article was published from the same study, data extraction was combined and reported as one study on the Microsoft Excel spreadsheet.

### Validity assessment

The quality assessment tool by Hoy and Brooks et al [24,25] was adapted and modified for use. The modified tool (S5 Table) evaluated both the external validity (selection and non-response bias) and internal validity (measurement bias and bias in relation with the analysis) of studies. An overall assessment of quality was made based on the reviewer's individual judgement which is in line with the Cochrane and the Grades of Recommendation, Assessment, Development and Evaluation (GRADE) processes [26,27].

### Data analysis

Where two or more studies reported a symptom, pooled proportions were determined using the "metaprop" command in Stata 15 (Stata corp, College Station, Texas, USA) [28,29]. Pooled estimates were calculated using the random effects meta-analysis model and Freeman-Tukey double arcsine transformation used to stabilize the variances. Pooled proportions and 95% "exact" confidence intervals (CI) were computed for individual studies by patient source category and overall across all studies for each symptom. Total variability between studies was quantified using the $I^2$ measure and the expected range of future true symptom prevalence across studies presented as a prediction interval [30]. Results from the meta-analysis were reported using forest plots for each symptom and further summarised in table form. The qualitative information on the characteristics of each symptom was summarised thematically. A map showing the distribution of studies included in this review was drawn using ArcGIS Arc-Map software version 10.5. Redlands, CA: Environmental Systems Institute, Inc., 2010.

## Results

Overall, 3765 articles were retrieved, of which 76 were assessed for full text eligibility and 32 included in data extraction and meta-analysis. The details of outputs from each database and reasons for exclusion are shown in Fig 1. Two pairs of articles reported data from the same studies and data from each pair was combined and reported as a single study.

### Study characteristics

Thirty-two publications from 30 studies reported on 21 outbreaks between 1933 and 2019 in 15 countries. The characteristics of included studies are shown in Table 1. All were from endemic countries (Fig 2) except the USA. Eight studies (26.7%) were of high quality, 17 (56.7%) moderate quality and 5 (16.7%) of low quality (S6 Table). Thirteen studies were case series, 12 cross-sectional and 5 cohort studies. The sample sizes varied from three to 683 patients, while ages ranged between two and 90 years.

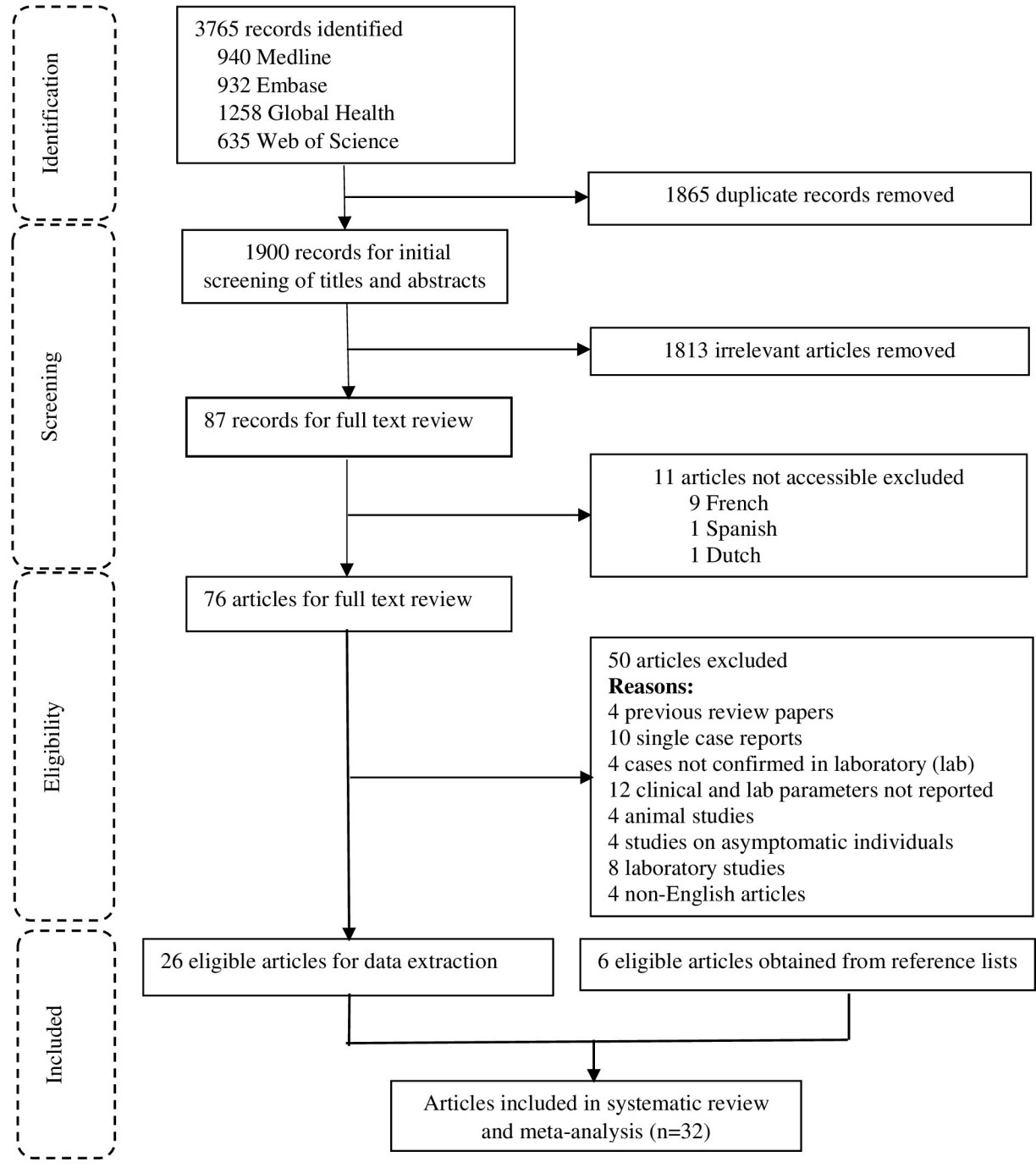

**Fig 1. Systematic review search flow chart.**

Most studies (25/30) were hospital based with 10 among in-patients, 11 both in-and out-patients, and 4 among out-patients. Studies published before 1972 reported disease exclusively among males, while thereafter, an increasing trend in reporting among women is observed (S1 Fig). RVF outbreaks occurred at any time of the year with two peaks between February-April and September-November (S2 Fig).

**Table 1. Main characteristics of studies included in the systematic review and meta-analysis.**

| Author; year of Publication (reference) | Design, Period (Country) | Subjects source | Test(s) done | Number screened | RVF +ve after screening N (%) | Enrolled N (%) | Age (years) | Main study aim |
|---|---|---|---|---|---|---|---|---|
| Adam; 2010[31] | Cross-sectional, Sep–Nov 2007 (Sudan) | In-patients | ELISA IgM | 18 | 18 (100%) | 18 M: 15 (83%) F: 3(17%) | Mean: 38.7 (sd: 14.4) Median: missing | Describe the manifestations, morbidity, and mortality related to the recent outbreak of RVF in central Sudan. |
| El Imam; 2009 [32] | Cohort, Sep 2007—Jan 2008 (Sudan) | In-patients | Not mentioned | 392 | Missing | 194 M: 145 (73%) F: 49 (25%) | Mean: 34 (sd: missing) Median: missing (range: 15–65) | Estimate the incidence of renal impairment, the demographics and modes of presentation as well as to determine the mortality rate related to renal impairment in patients with RVF |
| Baudin; 2016 [33] | Cross-sectional, Jun–Nov 2012 (Sudan) | In-patients | ELISA IgM PRNT rtPCR | 130 | 28 (22%) | 28 M: 0(0%) F: 28 (100%) | Mean: 27.8 (sd: 5.0) Median: missing (range: 17–37) | Determine which infectious agents were the cause of miscarriage in a cross-sectional study of febrile pregnant women who attended a hospital in Port Sudan, Sudan |
| Smithburn; 1949[34] | Case series, Dec 1944 – Apr 1948 (Uganda) | In-patients & out-patients/ laboratory workers | VNT MI&H MPT | 8 | 8 (100%) | 8 M: 8 (100%) F: 0(0%) | Mean: 30 (sd: 8.5) Median: 27 (range: 22–45) | Report these cases, together with certain incidental observations of significance |
| Henderson; 1972[35] | Cohort, April–May 1968 (Uganda) | In-patients & out-patients | CF VNT | 7 | 7 (100%) | 7 M: 6 (86%) F: 1(14%) | Mean: 28 (sd: 5.7) Median: 28 (range: 19–35) | Investigate the natural hosts and vectors of Rift Valley fever (RVF) virus during April and May 1968 outbreak affecting seven cases among persons living at Nakiwogo, Bunono and Lunyo on the outskirts of Entebbe near the East African Virus Research Institute (EAVRI) in Uganda |
| St. Maurice; 2016 & 2018 [36,37] | Case series, Mar–Jun 2016 (Uganda) | In-patients | ELISA IgM rtPCR | 3 | 3 (100%) | 3 M: 3 (100%) F: 0(0%) | Mean: missing Median: missing (range: 16–45) | Examine the physiologic consequences of RVFV infection in the human host using blood samples collected serially as part of clinical care. |
| Nguku; 2010 [20] | Cross-sectional, Nov 2006 – Mar 2007 (Kenya) | Community patients | ELISA IgM rt-PCR | 970 | 121 (12%) | 121 M: missing F: missing | Mean: missing Median: missing (range: 2–85) | Describes the magnitude and geographic scope of the outbreak and characterize epidemiologic, ecologic, and virologic features of the epidemic in Kenya |
| Kahlon; 2010 [17] | Case series, Dec 2006 –Feb 2007 (Kenya) | In-patients & out-patients | ELISA IgM rtPCR | 15 | 6 (40%) | 6 M: 3 (50%) F: 3(50%) | Mean: 31 (sd: 11.1) Median: 25 (range: 24–50) | Assess and fully evaluate both the early and late clinical course of acutely ill RVF patients |
| Anyangu; 2010 [38] | Cross-sectional, Jan–Mar 2007 (Kenya) | Community patients | ELISA IgM rtPCR | 861 | 202 (23%) | 202 M: missing F: missing | Mean: missing Median: missing | Determine risk factors associated with RVF infection, severe illness, and death |

*(Continued)*

**Table 1.** (Continued)

| Author; year of Publication (reference) | Design, Period (Country) | Subjects source | Test(s) done | Number screened | RVF +ve after screening N (%) | Enrolled N (%) | Age (years) | Main study aim |
|---|---|---|---|---|---|---|---|---|
| Abdel-Wahab; 1978[39] | Case series, Oct–Dec 1977 (Egypt) | In-patients & out-patients | CF VNT Histopathology | 13 | 13 (100%) | 13 M: 13 (100%) F: 0(0%) | Mean: missing Median: missing (range: 25–28) | Study several cases of infection, which occurred in Inshas, near Belbes, as well as two patients, admitted to Abbassia Fever Hospital in Cairo. |
| Laughlin; 1979 [40] and Siam; 1980[41] | Case series, Oct–Dec 1977 (Egypt) | In-patients | HAI MNT | Missing | 22 (100%) | 22 M: 13 (59%) F: 9(41%) | Mean: missing Median: missing (range: 5–52) | Report the clinical spectrum of human disease observed during the recent epidemic. |
| Madani; 2003 [5] | Cohort, Aug 2000 – Sep 2001 (Saudi Arabia) | In-patients | ELISA IgM, rtPCR IHC | 834 | 683 (82%) | 683 M: 565 (83%) F: 118 (17%) | Mean: 46.9 (sd: 19.4) Median: 50 (range: 10–90) | Study summarizes the epidemiological, clinical, and laboratory characteristics of this first confirmed occurrence of RVF outside Africa. |
| Mohammed Al-Hazmi; 2003 [21] | Cohort, Sep–Nov 2000 (Saudi Arabia) | In-patients | ELISA IgM ELISA IgG rtPCR Virologic typing | 165 | 165 (100%) | 165 M: 136 (82%) F: 29 (18%) | Mean: 47.5 (sd: missing) Median: 50 (range: 15–95) | Determine the clinical pattern of RVF, the frequency of its complications, and the associated case-fatality rates among patients in Saudi Arabia. |
| Ali Al-Hazmi; 2005[42] | Cross-sectional, Sep–Nov 2000 (Saudi Arabia) | In-patients & out-patients | ELISA IgM ELISA IgG | 329 | 319 (97%) | 143 M: 111 (78%) F:32(22%) | Mean: 53.2 (sd: missing) Median: missing (range: 14–80) | Determine the clinical pattern of ocular manifestations of RVF and to determine the outcome of ocular lesions during the follow-up period. |
| Kahiry; 2005 [43] | Cohort, Sep–Dec 2000 (Yemen) | In-patients & out-patients | ELISA IgM | 143 | 48 (34%) | 48 M: 25 (52%) F: 23 (48%) | Mean: 37.8 (sd: missing) Median: missing (range: 8–70) | Study the epidemiological and clinical pattern of positive RVF cases in Al-Zuhrah district—Hodiedah Governorate at the time of RVF epidemic in Yemen Sep—Dec 2000. |
| Swanepoel; 1979[44] | Case series, Feb–Jun 1977 (Zimbabwe) | In-patients & Out-patients | ID EM HAI | 45 | 43 (96%) | 43 M: missing F: missing | Mean: missing Median: missing | Report the occurrence of encephalitis, ocular complications and fatal haemorrhagic fever in Rhodesia |
| Lagare; 2019 [45] | Cross-sectional, Aug–Dec 2016 (Niger) | Community patients | ELISA IgM rt-PCR | 399 | 17 (4%) | 17 M: 6 (35%) F: 11 (65%) | Mean: 23 (sd: missing) Median: missing (range: 3–70) | Describe the outbreak and report the results of serological and molecular investigations of human and animal samples collected. |
| Joubert; 1951 [46] | Case series; Mar–May 1951 (South Africa) | Community patients | CF VNT | 33 | 23 (70%) | 23 M: missing F: missing | Mean: missing Median: missing | Investigation of this outbreak of Rift Valley fever was undertaken in the Bultfontein district of the Western Orange Free State. |

(*Continued*)

**Table 1.** (Continued)

| Author; year of Publication (reference) | Design, Period (Country) | Subjects source | Test(s) done | Number screened | RVF +ve after screening N (%) | Enrolled N (%) | Age (years) | Main study aim |
|---|---|---|---|---|---|---|---|---|
| Shrire; 1951[47] | Case series, Mar–Jun 1951 (South Africa) | Out-patients/farm workers | Serology | 6 | 6 (100%) | 6 M: 6 (100%) F: 0(0%) | Mean: 36 (sd: 8.2) Median: 34 (range: 28–50) | Describe five cases of macular exudates and one case of retinal detachment recently seen in my practice. All of these have been proved serologically. |
| Mundel; 1951 [48] | Case series, April 1951 (South Africa) | Out-patients/farm workers & veterinarians | MPT CF | 7 | 5 (71%) | 5 M: 5 (100%) F: 0(0%) | Mean: 41 (sd: 13.4) Median: 35 (range: 32–64) | Record an outbreak of human Rift Valley fever which originated at the farm Rietvlei, 10 miles south of the centre of Johannesburg |
| Van Velden; 1977[49] | Cross-sectional, Mar—May 1975 (South Africa) | In-patients | MI&H CF HAI EM | Missing | 17 | 17 M: 12 (71%) F: 5(29%) | Mean: missing Median: missing (range: 10–77) | Investigate the cause of relatively severe illness among 17 patients admitted to hospital in Bloemfontein |
| Archer; 2011 [50] | Cross-sectional, Feb–Mar 2008 (South Africa) | In-patients & outpatients | ELISA IgM rtPCR ISA Virus isolation | 53 | 8 (15%) | 8 M: 5 (63%) F: 3(37%) | Mean: Missing Median: missing (range: 20–29) | Report the subsequent outbreak of RVF in dairy farmers and farm workers, and the staff and students of a veterinary school. Investigated the prevalence of RVFV infection among them, their clinical presentation, and the risk factors associated with infection. |
| Jouan; 1988[51] | Cross-sectional, Oct 1987 (Mauritania) | In-patients & outpatients | ELISA IgM Virus isolation | Missing | 284 | 284 M: missing F: missing | Mean: missing Median: missing | Study the prevalence of recent infection and disease among Rosso residents. |
| Faye; 2007[52] | Case series, Sep–Dec 2003 (Mauritania) | In-patients & community patients | ELISA IgM Virus isolation (phylogenetics) | 98 | 17 (17%) | 17 M: missing F: missing | Mean: missing Median: 21 (range: 7–50) | Describe the results of a multidisciplinary investigation to determine the extent of outbreak and the key factors responsible for RVFV re-emergence in Mauritania. |
| Sow & Faye; 2014[53] | Cross-sectional, Sep–Nov 2012 (Mauritania) | Community patients | ELISA IgM ELISA IgG rtPCR | 288 | 41 (14%) | 41 M: 18 (44%) F: 23 (56%) | Mean: missing Median: 24 (range: 2–86) | Report the results of RVF investigation and laboratory findings from the 2012 RVF outbreak in Mauritania. |
| Boushab; 2016 [22] | Cross-sectional, Sep–Nov 2015 (Mauritania) | In-patients | ELISA IgM rtPCR | 57 | 31 (54%) | 31 M: 23 (74%) F: 8(26%) | Mean: 25 (sd: missing) Median: missing (range: 4–70) | Describe severe clinical signs and symptoms of Rift Valley Fever in southern Mauritania. |
| Gonzalez; 1987 [54] | Case series, 1971–1986 (Central African Republic) | Out-patients | Fluorescent antibody test | 3471 | 20 (0.6%) | 9 M: missing F: missing | Mean: missing Median: missing | The incidence of RVF, as determined by surveys of suspected human arboviral infections in the Central African Republic (CAR). |
| Sow; 2016[55] | Cross-sectional, Sep 2013 –Feb 2014 (Senegal) | In-patients | ELISA IgM rtPCR | 535 | 11 (2%) | 11 M: 7 (64%) F: 4(36%) | Mean: missing Median: 23 (range: 13–32) | Report multidisciplinary field investigations and laboratory findings in 3 regions of Senegal: Mbour, Linguere, and Kedougou. |

(*Continued*)

**Table 1.** (Continued)

| Author; year of Publication (reference) | Design, Period (Country) | Subjects source | Test(s) done | Number screened | RVF +ve after screening N (%) | Enrolled N (%) | Age (years) | Main study aim |
|---|---|---|---|---|---|---|---|---|
| Kitchen; 1934 [9] | Case series, Feb–Oct 1933 (USA) | Out-patients/ laboratory workers | VNT MI&H | 3 | 3 (100%) | 3 M: 3 (100%) F: 0(0%) | Mean: 28 (sd: 7.8) Median: 24 (range: 23–37) | The primary object of this report is to place on record three instances of accidental infection, contracted in the laboratory, with the virus of Rift Valley fever. |
| Francis; 1935[8] | Case series, Oct–Dec 1934 (USA) | In-patients & outpatients/ laboratory workers | MI&H | 3 | 3 (100%) | 3 M: 3 (100%) F: 0(0%) | Mean: missing Median: missing | Report deals with three cases of laboratory infection with Rift Valley fever in human individuals, in the first of which the source of the infection is obscure. |

CF, Complement fixation test; ELISA IgG, Enzyme linked immunosorbent Assay Immunoglobulin G; ELISA IgM, Enzyme linked immunosorbent Assay Immunoglobulin M; EM, Electron microscopy; F, Female; HAI, Haemaglutination inhibition test; ID, Agar gel immune diffusion tests; IHC, Immunohistochemistry of biopsy specimens; ISA, loop-mediated isothermal amplification assay; M, Male; MI&H, Mice inoculation & histological exam; MNT, Mouse neutralisation test; MPT, Mice protection test; N, Number; PRNT, Plaque reduction neutralisation test; rtPCR, real time reverse transcriptase Polymerase chain reaction; RVFV, Rift Valley fever virus; sd, standard deviation; USA, United States of America; VNT, Virus neutralisation test; %, Percentage; In-patients, subjects source in the study was hospital based patients requiring admission; Out-patients, subjects source in the study was hospital based patients requiring no admission; In-patients and Out-patients, subjects source in the study was both hospital based patients requiring admission and no admission and data collection in the included studies was combined; Community patients, subjects source in the study was non-hospital based patients found in the community or at home.

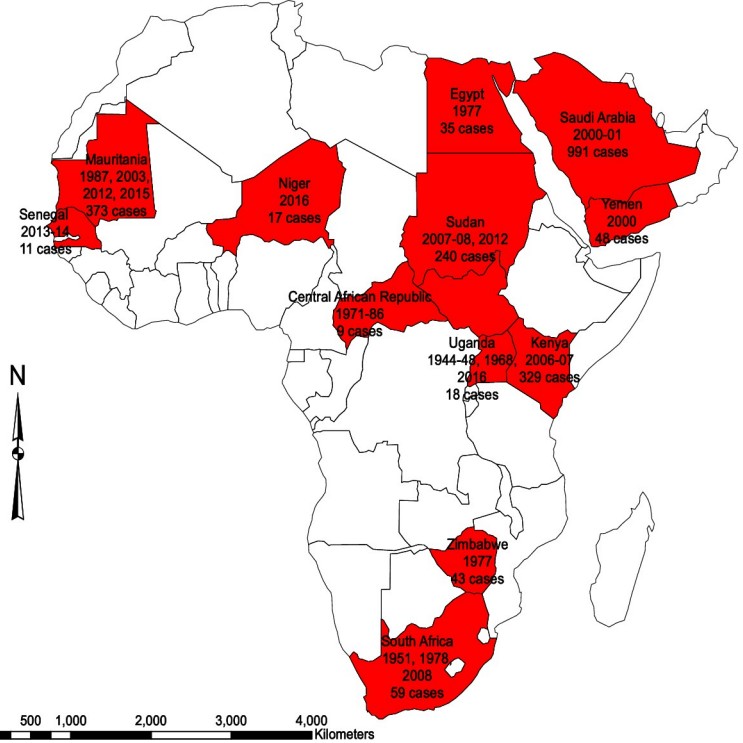

**Fig 2. Spatial-temporal distribution of RVF outbreaks and cases included in this systematic review.** Country source of studies, outbreak years and total number of cases included in this systematic review and meta-analysis. Figure created by authors. Basemap source: https://www.naturalearthdata.com/downloads/50m-cultural-vectors/50m-admin-0-countries-2/.

## Clinical syndromes and use of case definitions

Six different previously described RVF clinical syndromes were reported. Seven studies reported on the general febrile (influenza-like or dengue-like, uncomplicated RVF or classic) syndrome [22,35,40,49,51,53,54], one study on gastrointestinal [5], two studies on hepatic (icteric) [42,51], two studies neurological (encephalitis or meningo-encephalitis) [22,51], five studies haemorrhagic [22,42,43,51,54], and two studies on visual (ocular or retinitis or blurred vision) [41,43] syndromes. The other manifestations not previously defined were categorised as renal [32], cardio-pulmonary [44], and obstetric syndromes [33]. Death was reported in 16 out of the 32 included studies. S7 Table shows the studies that used a case definition to identify patients, the clinical syndromes covered by the case definition versus manifested by patients in the respective outbreaks. Sixteen studies did not report or use a case definition in identifying RVF patients and those that had one, the case definitions included symptoms of up to 6 RVF syndromes. A review of all studies revealed symptoms of 9 clinical syndromes in addition to death. Some studies reporting from the same outbreaks in Kenya [17,20,38] and the Arabian peninsula [5,43], used a variety of case definitions.

## RVF clinical manifestations

Table 2 shows the pooled proportions for RVF symptoms estimated using the random effects meta-analysis and Fig 3, the forest plots of the most common and precisely measured symptoms from each syndrome. The rest of the forest plots corresponding to the information summarised in Table 2 for the pooled proportions of RVF symptoms are indicated in S3–S10 Figs. S8 Table is a summary table describing the characteristics of the different RVF clinical symptoms. The proportion of patients with symptoms for which a pooled estimate could not be calculated are indicated in S9 Table.

**General febrile/influenza-like syndrome.** Most RVF cases with this syndrome presented with fever (81%, 95% CI 69–91; [26 studies, 1286 patients]), headache (71%, 51–91; [21, 413]), arthralgia (68%, 41–91; [16, 260]) and myalgia (66%, 40–89; [15, 211]). Patients also reported chills (49%, 20–78; [9, 89]), malaise (47%, 26–72; [8, 51]), backache (37%, 0–87; [6, 21]) and fatigue (32%, 0–89; [3, 16]). Other manifestations for which a pooled estimate could not be estimated included joint stiffness, dehydration, weight loss, prostration, nasal congestion and a sore throat (S9 Table). The symptoms under this syndrome were of varied intensity and period of onset. There was a sudden onset of very high fever [39,40,44,46], severe headache [34,39,40,44,46,49], myalgia [34,39,40,49], arthralgia [34,39,46,49], and backache [49], associated with chills [34,49], rigors [34,44], malaise [33,34,40], and easy fatiguegability [9]. In some instances the onset of fever, headache, myalgia, arthralgia and backache was slow with a mild intensity [34,53] or completely absent [17,34]. The fever was recurring [40] or had a biphasic pattern [8,9,33,47,48], with a saddle-back temperature curve [34,48]. Specifically, patients experienced a high fever exceeding 39˚C for 2 to 4 days, followed by a return to normal temperature for 1 to 2 days and thereafter a relapse for 1 to 3 days. The recrudescence of fever coincided with the worsening of other manifestations [34,46,47]. Patients complained of a throbbing headache that was diffuse or frontal [9,17,39], associated with retro-orbital pain [44]. They also experienced severe or backbreaking myalgias [40] of shooting nature [8] and muscular weakness which persisted longer than the fever [9]. The excruciating arthralgia and joint stiffness [34] mainly affected the proximal large joints of the knee [46], elbow and shoulder with no tenderness, effusion or pain on active and passive joint movement [17].

**Gastrointestinal syndrome.** This was characterised by epigastric discomfort (58%, 30–84; [2, 8]), vomiting (43%, 15–74; [4, 288]), nausea (38%, 12–67; [6, 325]), nausea and vomiting

**Table 2. Pooled proportions of RVF clinical manifestations estimated using random effects meta-analysis.**

| Syndromes and symptoms | Patient source | | | | | | | | | |
|---|---|---|---|---|---|---|---|---|---|---|
| | In-patients | | Out-patients | | In- and Out-patients | | Community patients | | Overall | |
| | n | % (95% CI) | n | % (95% CI) | n | % (95% CI) | n | % (95% CI) | n | % (95% CI) |
| **General febrile/influenza-like syndrome** | | | | | | | | | | |
| Fever | 863 | 78 (57; 95) | 16 | 79 (45; 100) | 241 | 86 (59; 100) | 166 | 77 (61; 90) | 1286 | 81 (69; 91) |
| Headache | 127 | 54 (20; 85) | 8 | 100 (77; 100) | 128 | 86 (62; 100) | 150 | 55 (16; 91) | 413 | 73 (51; 91) |
| Arthralgia | 34 | 41 (5; 81) | 8 | 100 (77; 100) | 78 | 78 (45; 99) | 140 | 67 (39; 89) | 260 | 68 (41; 91) |
| Myalgia | 116 | 51 (15; 86) | 7 | 93 (59; 100) | 57 | 88 (63; 100) | 31 | 45 (33; 58) | 211 | 66 (40; 89) |
| Chills | 43 | 22 (16; 29) | 4 | 55 (17; 91) | 42 | 49 (12; 86) | - | - | 89 | 49 (20; 78) |
| Malaise | 29 | 55 (29; 80) | 1 | 20 (1; 72)* | 21 | 48 (15; 81) | - | - | 51 | 47 (26; 72) |
| Backache | 2 | 1 (0; 4)* | 4 | 50 (13; 87) | 15 | 58 (19; 93) | - | - | 21 | 37 (0; 87) |
| Fatigue | 12 | 6 (3; 11)* | 3 | 100 (29; 100)* | 1 | 17 (0; 64)* | - | - | 16 | 32 (0; 89) |
| Lethargy | 42 | 7 (5; 10) | 4 | 80 (28; 99)* | 2 | 14 (0; 40) | 2 | 9 (1; 28)* | 50 | 18 (5; 35) |
| Rash | 10 | 36 (19; 56)* | - | - | 1 | 2 (0; 11)* | - | - | 11 | 11 (4; 19) |
| **Gastrointestinal syndrome** | | | | | | | | | | |
| Epigastric discomfort | - | - | - | - | 8 | 58 (30; 84) | - | - | 8 | 58 (30; 84) |
| Vomiting | 281 | 53 (48; 58) | - | - | 7 | 27 (10; 49) | - | - | 288 | 43 (15; 74) |
| Nausea | 315 | 59 (55; 64)* | 1 | 33 (1; 91)* | 9 | 31 (3; 68) | - | - | 325 | 38 (12; 67) |
| Nausea and vomiting | 169 | 42 (3; 88) | 2 | 40 (5; 85) | 24 | 24 (4; 52) | 53 | 44 (35; 53) | 248 | 35 (12; 62) |
| Anorexia | 23 | 29 (0; 58) | 4 | 80 (28; 99)* | 12 | 49 (1; 99) | 1 | 4 (0; 22)* | 40 | 31 (9; 58) |
| Abdominal pain | 288 | 23 (4; 49) | 1 | 33 (1; 91) | 25 | 18 (4; 39) | - | - | 314 | 21 (7; 38) |
| Diarrhoea | 207 | 21 (7; 39) | 1 | 33 (1; 91)* | 13 | 12 (6; 20) | - | - | 221 | 18 (7; 32) |
| Constipation | - | - | - | - | 2 | 25 (3; 65) | 1 | 4 (0; 22) | 3 | 8 (0; 21) |
| **Hepatic syndrome** | | | | | | | | | | |
| Liver failure | 177 | 36 (9; 69) | - | - | - | - | - | - | 177 | 36 (9; 69) |
| Right hypochondriac tenderness | - | - | - | - | 34 | 46 (11; 83) | 1 | 4 (0; 22) | 35 | 31 (1; 75) |
| Hepatomegaly | 25 | 13 (8; 18) | - | - | 31 | 64 (49; 78) | - | - | 56 | 29 (3; 64) |
| Jaundice | 216 | 32 (17; 50) | - | - | 152 | 21 (5; 43) | 25 | 17 (11; 25) | 393 | 26 (16; 36) |
| Splenomegaly | 25 | 13 (9; 19) | - | - | 2 | 2 (0; 9) | - | - | 27 | 11 (4; 19) |
| Elevated AST | 656 | 96 (94; 97) | - | - | 4 | 100 (40; 100) | - | - | 660 | 97 (87; 100) |
| Elevated ALT | 625 | 94 (93; 96) | - | - | - | - | - | - | 625 | 94 (93; 96) |
| Elevated LDH | 304 | 52 (36; 68) | - | - | - | - | - | - | 304 | 52 (36; 68) |
| **Renal syndrome** | | | | | | | | | | |
| Renal failure | 327 | 41 (23; 59) | - | - | - | - | - | - | 327 | 41 (23; 59) |
| Elevated creatinine | 184 | 33 (29; 37) | - | - | - | - | - | - | 184 | 33 (29; 37) |
| **Neurological syndrome** | | | | | | | | | | |
| Dizziness | - | - | 3 | 100 (29; 100)* | 6 | 55 (22; 86) | 1 | 4 (0; 22)* | 10 | 46 (1; 95) |
| Delirium | - | - | 3 | 37 (12; 74) | 5 | 83 (36; 100)* | 5 | 22 (7; 44)* | 13 | 42 (12; 74) |
| Insomnia | - | - | 1 | 33 (1; 91)* | 1 | 33 (1; 91)* | - | - | 2 | 33 (0; 79) |
| CNS symptoms/encephalitis | 145 | 29 (15; 45) | - | - | 28 | 10 (2; 23) | - | - | 173 | 21 (11; 33) |
| Hyperaesthesia | - | - | - | - | 2 | 13 (0; 35) | - | - | 2 | 13 (0; 35) |
| Coma | 47 | 11 (3; 21) | - | - | 5 | 5 (1; 11) | - | - | 52 | 9 (4; 16) |
| Vertigo | 19 | 3 (1; 4) | - | - | 1 | 13 (0; 53)* | - | - | 20 | 9 (0; 31) |
| Meningismus | 12 | 10 (0; 38) | - | - | 2 | 3 (0; 10) | - | - | 14 | 7 (0; 20) |
| Confusion | 54 | 8 (3; 15) | - | - | 3 | 7 (0; 34) | - | - | 57 | 5 (1; 11) |
| Disorientation | 32 | 5 (3; 8) | - | - | - | - | - | - | 32 | 5 (3; 8) |
| Hallucinations | 7 | 0 (0; 1) | - | - | - | - | - | - | 7 | 0 (0; 1) |
| Ataxia | 3 | 1 (0; 2)* | - | - | 1 | 33 (1; 91) | - | - | 4 | 0 (0; 0) |

(*Continued*)

**Table 2.** (Continued)

| Syndromes and symptoms | Patient source | | | | | | | | | |
|---|---|---|---|---|---|---|---|---|---|---|
| | In-patients | | Out-patients | | In- and Out-patients | | Community patients | | Overall | |
| | n | % (95% CI) | n | % (95% CI) | n | % (95% CI) | n | % (95% CI) | n | % (95% CI) |
| Choreiform movements | 4 | 0 (0; 1) | - | - | - | - | - | - | 4 | 0 (0; 1) |
| Hemiparesis | 3 | 0 (0; 0) | - | - | - | - | - | - | 3 | 0 (0; 0) |
| Locked-in-syndrome | 2 | 0 (0; 0) | - | - | - | - | - | - | 2 | 0 (0; 0) |
| **Haemorrhagic syndrome** | | | | | | | | | | |
| Pallor | 84 | 44 (37; 51) | - | - | 1 | 2 (0; 11) | - | - | 85 | 40 (3; 87) |
| Haemorrhagic disease | 121 | 31 (13; 52) | 4 | 44 (14; 79)* | 80 | 15 (6; 25) | 72 | 31 (19; 44) | 277 | 26 (17; 36) |
| Epistaxis | 84 | 35 (4; 74) | 1 | 20 (1; 72) | 11 | 13 (5; 24) | 23 | 14 (9; 20) | 119 | 22 (7; 40) |
| Haematemesis | 102 | 16 (4; 34) | - | - | 6 | 6 (1; 13) | 20 | 12 (7; 17) | 128 | 12 (5; 21) |
| Melena | 46 | 11 (1; 27) | - | - | 4 | 6 (1; 15) | 14 | 12 (6; 19)* | 64 | 10 (3; 19) |
| Shock | 23 | 12 (7; 17) | - | - | 3 | 5 (0; 13) | - | - | 26 | 10 (6; 14) |
| Bleeding gums | 57 | 11 (0; 30) | - | - | 6 | 6 (1; 13) | 13 | 11 (6; 18)* | 76 | 9 (2; 19) |
| Sub-conjunctival haemorrhage | 26 | 3 (2; 4) | - | - | 1 | 17 (0; 64)* | - | - | 27 | 4 (0; 18) |
| Petechiae | 24 | 3 (1; 6) | - | - | 1 | 8 (0; 36)* | - | - | 25 | 3 (1; 6) |
| Haematochezia | 5 | 0 (0; 1) | - | - | 3 | 6 (1; 17)* | - | - | 8 | 3 (0; 11) |
| Uterovaginal bleeding | 31 | 2 (0; 8) | - | - | - | - | - | - | 31 | 2 (0; 8) |
| Ecchymoses | 12 | 3 (2; 5) | - | - | 2 | 3 (0; 10) | - | - | 24 | 2 (1; 5) |
| Macular/purpura rash | 12 | 1 (0; 2) | - | - | 1 | 2 (0; 12) | - | - | 13 | 2 (0; 6) |
| Haemoptysis | 2 | 1 (0; 2) | - | - | - | - | - | - | 2 | 1 (0; 2) |
| Thrombocytopenia | 260 | 47 (31; 63) | | | | | | | 260 | 47 (31; 63) |
| Anaemia (low Hb) | 103 | 15 (6; 27) | - | - | - | - | - | - | 103 | 15 (6; 27) |
| **Visual syndrome** | | | | | | | | | | |
| Injected conjunctiva | - | - | 7 | 91 (55; 100) | 18 | 37 (0; 91) | - | - | 25 | 55 (9; 96) |
| Eye pain | 12 | 71 (44; 90)* | - | - | 22 | 42 (27; 58) | - | - | 34 | 53 (31; 74) |
| Retro-orbital pain | 7 | 32 (14; 55)* | 1 | 33 (1; 91)* | 11 | 85 (55; 98)* | - | - | 19 | 53 (13; 92) |
| Blurred or partial blindness | 17 | 8 (1; 22) | 6 | 100 (54; 100)* | 156 | 20 (2; 48) | 46 | 32 (24; 40) | 225 | 24 (7; 45) |
| Photophobia | 1 | 33 (1; 91)* | 1 | 20 (1; 72)* | 21 | 15 (0; 51) | - | - | 23 | 17 (0; 46) |
| Retinitis | 17 | 7 (3; 12) | - | - | - | - | - | - | 17 | 7 (3; 12) |
| **Cardio-pulmonary syndrome** | | | | | | | | | | |
| Syncope | 1 | 17 (0;64) | 2 | 40 (5; 85) | - | - | - | - | 3 | 27 (3; 59) |
| Throat swelling | | | - | - | 5 | 5 (0; 16) | - | - | 5 | 5 (0; 16) |
| Cough | 8 | 3 (1; 7)* | 1 | 20 (1; 72)* | 2 | 6 (0; 25) | - | - | 11 | 4 (0; 17) |
| **Other—death** | | | | | | | | | | |
| Death | 258 | 30 (18; 44) | 3 | 33 (7; 70)* | 37 | 8 (5; 11) | 30 | 20 (3; 45) | 328 | 21 (14; 29) |

*, based on a single study where pooled estimate could not be calculated; n, Number of patients; %, percentage; CI, Confidence interval; CNS, Central nervous system; AST, aspartate aminotransferase; ALT, Alanine aminotransferase; LDH, Lactate dehydrogenase; Hb, Haemoglobin; In-patients, subjects source in the study was hospital based patients requiring admission; Out-patients, subjects source in the study was hospital based patients requiring no admission; In-patients and Out-patients, subjects source in the study was both hospital based patients requiring admission and no admission and data collection in the included studies was combined; Community patients, subjects source in the study was non-hospital based patients found in the community or at home.

(35%, 12–62; [10, 248]), anorexia (31%, 9–58; [8, 40]), abdominal pain (21%, 7–38; [11, 314]) and diarrhoea (18%, 7–32; [9, 221]). In addition, patients presented with a coated tongue, dysphagia and/or odynophagia. The onset of anorexia, nausea and vomiting were sudden and persisted during recovery [17,34]. Patients poorly localised the abdominal pain [23] or reported vague epigastric discomfort early in the illness [17].

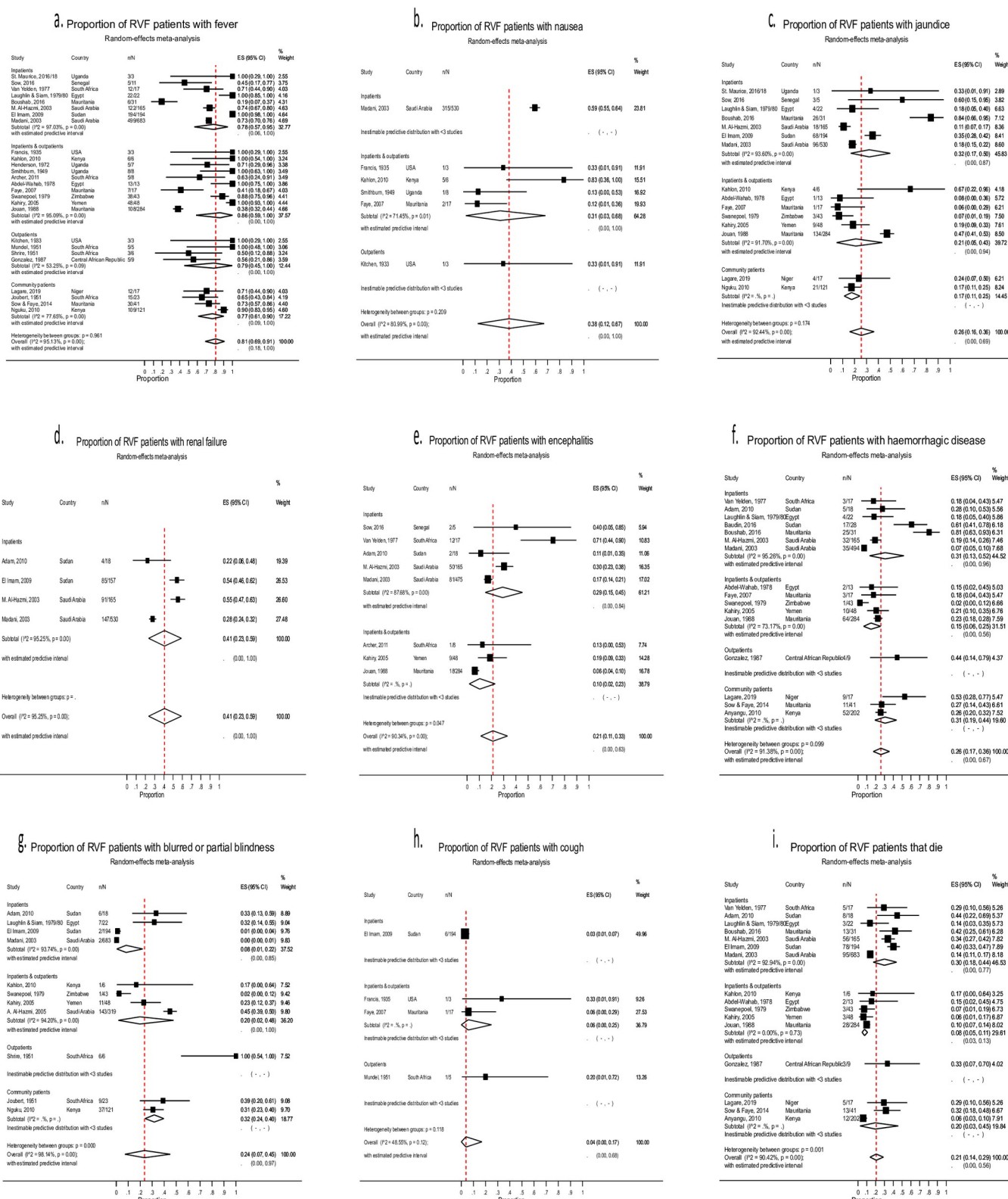

This is the Fig 3 legend

a – proportion of RVF patients with fever; b – proportion of RVF patients with nausea; c – proportion of RVF patients with jaundice; d – proportion of RVF patients with renal failure; e – proportion of RVF patients with encephalitis; f – proportion of RVF patients with haemorrhagic disease; g – proportion of RVF patients with blurred or partial blindness; h – proportion of RVF patients with cough; i – proportion of RVF patients that die; n – number of patients with the sign or symptom; N – total number of patients in the study assessed for sign or symptom; % – percentage; ES (95% CI) – estimated 95% confidence interval; % weight – percentage weight of the study calculated from random effects meta-analysis; I² – chi-square value; p – p-value; Inpatients - subjects source in the study was hospital based patients requiring admission; Outpatients – subjects source in the study was hospital based patients requiring no admission; Inpatients and outpatients - subjects source in the study was both hospital based patients requiring admission and no admission and data collection in the included studies was combined; Community patients - subjects source in the study was non-hospital based patients found in the community or at home.

**Fig 3. Forest plots for the most common symptoms from each RVF syndrome.** [5,8,9,17,20–22,31–55].

**Hepatic syndrome.** Hepatic manifestations included liver failure (36%, 9–69; [4, 177]), right hypochondriac tenderness (31%, 1–75; [4, 35]), hepatomegaly (29%, 3–64; [4, 56]), jaundice (26%, 16–36; [15, 393]), AST elevation (97%, 97–100; [3, 660]) and raised ALT (94%, 93–96; [2, 625]) and LDH (52%, 36–68; [3, 304]). Nearly all patients with liver failure required in-patient care. The hepatomegaly was usually mild [17,40,44], and the jaundice occurred between the first and third week following infection [17,40]. Other laboratory manifestations of liver damage included hyperbilirubinemia, elevated alkaline phosphatase, and prolonged prothrombin time [40,41].

**Renal syndrome.** This presented mainly as renal failure (41%, 23–59; [4, 327]) and was entirely acute [32] and nearly all cases required in-patients admission. Signs of renal failure included oliguria, pedal and/or pulmonary oedema [32]. Thirty-three percent (95% CI 29–37; [2, 184]) of patients developed elevated serum creatinine levels.

**Neurological syndrome.** Neurological manifestations included encephalitis (21%, 11–33; [8, 173]), dizziness (46%, 1–95; [4, 10]), delirium (42%, 12–74; [4, 13]) and insomnia (33%, 0–79; [2, 2]). In addition, patients presented with hyperaesthesia (13%, 0–35; [2, 2]), vertigo (9%, 0–31; [3, 20]), meningismus (7%, 0–20; [5, 14]), confusion (5%, 1–11; [6, 57]), disorientation (5%, 3–8; [2, 32]), and coma (9%, 4–16; [7, 52]). Other symptoms of meningo-encephalitis included drowsiness, irritability, lack of gustatory discrimination, tremors, amnesia, asthenia, neck pain, neck stiffness, decerebrate posturing, hypersalivation, and visual hallucinations. Their relative non-pooled proportions are indicated in S9 Table. The delirium coincided with the peak of clinical severity and/or onset of haemorrhage [17] and patients experienced a feeling of numbness along the spine and legs [34]. The cerebrospinal fluid (CSF) of patients with meningoencephalitis showed pleocytosis but normal CSF glucose and protein concentrations. CSF cell counts ranged between 20 and 600 cells/mm$^3$, predominantly lymphocytes [40].

**Haemorrhagic syndrome.** Patients experienced mild to severe forms of haemorrhage. In 26% (95% CI 17–36; [16, 277]) of patients, symptoms were reported as haemorrhagic disease without specifying the detailed manifestations. Where proportions of specific haemorrhagic symptoms were reported, these included pallor (40%, 3–87; [3, 85]), epistaxis (22%, 7–40; [10, 119]), hematemesis (12%, 5–21; [11, 128]), melena (10%, 3–19; [8, 64]), and gum bleeding (9%, 2–19; [9, 76]). Shock occurred in 10% (95% CI 6–14; [4, 26]) of patients. Other haemorrhagic manifestations included sub-conjunctival haemorrhage (4%, 0–18; [3, 27]), petechiae (3%, 1–6 [5, 25]), ecchymoses (2%, 1–5; [4, 24]), macular/purpura rash (2%, 0–6; [3, 13]), haematochezia (3%, 0–11; [3, 8]), haemoptysis (1%, 0–2; [2, 2]), and utero-vaginal bleeding (2%, 0–8; [4, 31]). Haemorrhagic symptoms from single studies for which a pooled estimate could not be calculated included hematuria, rectal bleeding, bleeding from puncture sites and hypotension (S9 Table). The haemorrhage appeared within 2 to 4 days [40] and epistaxis was persistent [8]. The maculopurpura rashes [40,44] and petechiae [40,41] were mild and generalised. Disseminated intravascular coagulopathy [23] presented with severe or profuse bleeding [49] from the nose, gums, skin and gastrointestinal tract (GIT) [17]. GIT bleeding was reported as frank blood in stool [17] or old blood with the appearance of coffee grounds [44], which led to anaemia [49]. The commonest laboratory manifestations of haemorrhage included thrombocytopenia (47%, 31–63; [3, 260]), and low haemoglobin (15%, 6–27; [3, 103]). Patients presented with an initial leucocytosis followed by leucopenia, then returned to normal within a week [40,41].

### Obstetric syndrome

One eligible study reported on the obstetric manifestations of RVF which included non-pooled proportions of miscarriage and preterm delivery in 15 (54%) and 1 (3%) of pregnant women, respectively. Those with miscarriage experienced severe haemorrhage [33]. Only 12 (43%) had full term pregnancies and a normal delivery.

**Visual syndrome.** This was characterised by blurring of vision or partial blindness (24%, 7–45; [11, 225]), an injected conjunctiva (55%, 9–96 [6, 25]), retro-orbital pain (53%, 13–92 [3, 19]), eye pain (53%, 31–74; [3, 34]), photophobia (17%, 0–46 [5, 23]), and retinitis (7%, 3–12; [2, 17]). The non-pooled ocular manifestation included red eyes (43.7%), retinal haemorrhages (5%), and complete blindness (1.2%) (S9 Table). The onset of eye pain and diminished visual acuity were sudden, and respectively, presented as pain behind the eyeball and variation in light perception and finger counting. Symptoms developed within 2–7 days and persisted for 10–15 days after onset of illness [40]. Severe visual impairment developed after 4 weeks following onset of illness, and presented as blurred vision, fogginess or haze, floating black spots or gap within visual fields [17,47]. Fundoscopy revealed macular, paramacular or extra-macular exudates [40,41] which appeared as multiple yellow plaques of variable sizes [17,47] with mild erythema at the borders [17] and associated haemorrhage [40,41,47] or retinal detachment [47]. In addition, severe uveitis with keratic precipitates and vitreous haze, vasculitis with peripapillary choroidal ischaemia and infarction were seen. In severe cases vessels were sheathed and occluded leading to optic atrophy [40,41]. Symptoms largely resolved within 2 weeks [17], though in some persisted for 3 months or more [46].

**Cardio-pulmonary syndrome.** Symptoms suggestive of this syndrome were rare. Patients presented with cough (4%, 0–17; [4, 11]) and syncope (27%, 3–59 [2, 3]). Non-pooled manifestations included chest pain (13%), dyspnoea (10%), hiccups (6%), myocarditis (5%), and pneumonia (2%) (S9 Table). One fatal case presented with interstitial pneumonitis typical of viral pneumonia.

**Other manifestations–death.** Death occurred in 21% (95% CI 14–29; [16, 328]) of cases, mostly among those that sought care from the hospital. The proportion was high (30%, 18–44; [7, 258]) among in-patients. The most common causes of death were isolated acute hepatic or renal failure, compound hepatorenal impairment, and shock within the first week of illness [31,32,40,41].

**RVF clinical case definition.** A suspected RVF case can be clinically defined as any human subject presenting with the general febrile/influenza-like and hepatic syndromes whether or not associated with the gastrointestinal, renal, neurological, haemorrhagic, obstetric, visual, and cardio-pulmonary syndromes and death. The detailed manifestations in patients would be as shown in Table 3 below.

**Co-infections.** A number of studies reported on the presence of concurrent infections among RVF patients. In one study, 4 (13%) RVF confirmed patients had hepatitis B surface antibodies [22]. Two studies reported RVF-malaria co-infections: in one, 2 (12%) patients were positive for malaria [52], while in the other 2 (25%) had malaria and 1 (13%) herpes simplex [34]. Chikungunya and dengue co-infections (positive PCR) were reported, respectively, in 8 (29%) and 9 (32%) RVF patients in Sudan [33]. In a study where two deaths occurred, the autopsy of one patient showed infection with both schistosomiasis and systemic fungal infection 1 (8%). The schistosomiasis infection was thought to have contributed to the severe liver necrosis while the systemic fungal infection occurred as a terminal complication [39].

## Discussion

This review identified signs and symptoms belonging to 6 previous, explicitly described syndromes including the general febrile or influenza-like, gastrointestinal, hepatic, neurological,

**Table 3. Proposed RVF clinical case definition summarised from results of systematic review and meta-analysis.**

| Patients' classical presentation |
| --- |
| **1. General febrile/influenza-like syndrome**<br>**Common:** fever, headache, arthralgia, myalgia, chills, malaise, backache, fatigue, lethargy, rash.<br>**Rare:** weight loss, prostration, nasal congestion, dehydration, joint stiffness, sore throat |
| **2. Hepatic syndrome**<br>**Common:** elevated ALT, elevated AST, elevated LDH, jaundice, right hypochondriac tenderness, hepatomegaly, liver failure, splenomegaly<br>**Rare:** ascites, elevated gamma glutamyl transferase (GGT) |
| **And/or any of the following forms** |
| **3. Gastrointestinal syndrome**<br>**Common:** epigastric discomfort, vomiting, nausea, anorexia, abdominal pain, diarrhoea, constipation.<br>**Rare:** coated tongue, dysphagia, odynophagia |
| **4. Renal syndrome**<br>**Common:** renal failure, elevated creatinine<br>**Rare:** elevated creatine phosphokinase (CPK) |
| **5. Neurological syndrome**<br>**Common:** dizziness, delirium, insomnia, CNS symptoms/encephalitis, hyperaesthesia, coma, vertigo, meningismus, confusion, disorientation, hallucinations, ataxia, choreiform movements, hemiparesis, locked-in-syndrome<br>**Rare:** drowsiness, hypersalivation, neck pain, lack of gustatory discrimination, asthenia, visual hallucinations, irritability, decerebrate posturing, neck stiffness, tremors, amnesia |
| **6. Haemorrhagic syndrome**<br>**Common:** pallor, haemorrhagic disease, epistaxis, haematemesis, melena, shock, bleeding gums, sub-conjunctival haemorrhage, petechiae, haematochezia, uterovaginal bleeding, ecchymoses, macular/purpura rash, haemoptysis, thrombocytopenia, anaemia (low Hb)<br>**Rare:** hypotension, haemorrhagic meningo-encephalitis, disseminated intravascular coagulation, haematuria, rectal bleeding, bleeding from puncture sites |
| **7. Visual syndrome**<br>**Common:** injected conjunctiva, eye pain, retro-orbital pain, blurred or partial blindness, photophobia, retinitis<br>**Rare:** red eye, retinal haemorrhage, complete blindness |
| **8. Obstetric syndrome**<br>**Rare:** abortion / miscarriage, pre-term delivery |
| **9. Cardio-pulmonary syndrome**<br>**Common:** syncope, throat swelling, cough<br>**Rare:** chest pain, dyspnoea, hiccup, myocarditis, pneumonia |
| **10. Death**—common |

Legend: Common–two or more studies reported on the sign/symptom and a pooled prevalence estimate could be calculated using random effects meta-analysis; Rare–a single study reported on the sign/symptom and a pooled prevalence estimate could not be calculated using random effects meta-analysis.

haemorrhagic, and visual syndromes [17,21,22,32,42,43,51,53,54]. Although symptoms of kidney, heart/lung, and pregnancy injury were previously documented, we did not find their corresponding syndromic terms and have thus assigned them to renal, cardio-pulmonary and obstetric syndromes respectively. The general febrile and hepatic syndromes were the most common presentation and 81% and nearly 100% of patients developed fever and elevation in liver transaminases respectively. A similar clinical picture was described in animals by Daubney et al in 1931 [2]. The dominancy of manifestations under these two syndromes characterised by influenza-like symptoms and liver abnormalities clearly underscores these as cardinal in defining a clinical RVF case. Despite this, the RVF virus infects nearly all body tissues producing a broad spectrum of human disease.

The varied RVF clinical picture is mainly attributed to the nature of the infecting virus strain as well as host factors [56,57]. Some animal studies have shown that RVF strains have diverse tropism for body tissues [34,58]. Although RVF is pantropic in its natural form [59], hepatotropic viruses predominate in most epizootics/epidemics [60]. Neurotropic viruses have been produced in the laboratory through several intracerebral passages in mice though these

have not been demonstrated as naturally occurring in humans. One accidental human infection in the laboratory with a neurotropic virus had a hepatotropic virus recovered instead [34]. Whilst no human studies have been conducted to conclusively study the association between RVF tropic nature and its symptomatology, it is possible that this multifarious nature could partly be responsible for the dominance of certain syndromes in some outbreaks [5,21,31]. Without phylogenetics capacity, it is difficult to tease out the virus strains responsible for many of the symptoms since a single outbreak can be caused by several viral strains [52,61].

Another viral virulence factor is the inherent nature of the RVF viral genome. RVF genome segments of more virulent strains can undergo genetic re-assortment with others of the same or lower virulence resulting into lineages with amplified virulence and a more severe disease [7]. One human case of a naturally attenuated RVF strain (clone 13) exhibiting a 69% deletion in the S genome segment in Central African Republic was associated with mild disease [62]. The S segment codes for the non-structural NSs protein which is a major virulence factor in RVF pathogenesis. NSs inhibits host cell interferon beta (IFN-1β) messenger ribonucleic acid (RNA) synthesis which is responsible for early clearance of the virus before the humoral response sets in [63]. The failure in early viral clearance may result in a dysregulated cytokine release leading to multiple organ injury and symptomatology [37]. The NSs protein also forms filamentous structures that interact with nuclear chromatin to cause chromosomal segregation and cohesion defects [64,65] and this is inferred to be partly the mechanism for the congenital anomalies and abortion storms in animals [60]. In fact, a recent study by Oymans et al found that the RVF virus directly invades sheep placental epithelial cells and fetal trophoblasts causing placental tissue necrosis, haemorrhage and subsequently induction of abortion. The virus was also found to replicate efficiently in human placental explants [66]. Although a case of human congenital anomalies due to RVF has not been documented, RVF vertical transmission in humans is known to occur [67,68]. Congenital anomalies following RVF infection are a common feature of disease presentation in animals [69,70] and another study by McMillen and colleagues showed that the RVF virus exhibits high tropism for human placental tissue and could be a more dangerous pathogen to the fetus than the Zika virus [71]. In this review, we observed an increased reporting of disease among women yet found only one eligible study that reported on the increased frequency of abortions among women [33]. Abortion storms are a major marker of RVF epizootics in domestic ruminants [2,4]. Studies are lacking on the association between RVF and human miscarriages as well as congenital anomalies.

Host factors associated with RVF clinical diversity include single nucleotide polymorphisms (SNP) in genes that code target cell surface molecules such as Toll-like Receptors (TLR) that detect pathogen associated molecular pattern (PAMPs), and molecules involved in the downstream signalling pathway and inflammatory mediators that play a major role in viral clearance via several pathways [72,73]. Thus, some individuals are able to elicit an early, rapid and effective cell mediated response leading to a subclinical or mild illness. SNP in TLR3, TLR7, TLR8, MyD88, TRIF, MAVS and RIG-1 are consistently associated with severe RVF symptoms in patients [72]. SNP in the genes that code TLR, human major histocompatibility complex (MCH), intracellular signalling pathway proteins and effector molecules such as cytokines have also been observed to cause a variation in response to vaccines [74].

Generally, effective neutralising antibodies appear within 4–6 days following RVF infection leading to the resolution of symptoms [11,75]. Unfortunately, severe RVF disease forms are reported to occur in the presence of an effective humoral response [57,63] and in one study has been inconclusively associated with RVF induced autoimmune retinitis [76]. This autoimmune mechanism could presumably be responsible for other late manifestations in other tissues. On the other hand, it could be that the magnitude and avidity of antibodies produced is variable (i.e. low and ineffective) such that the virus persists in some body tissues longer than

is currently known causing more injury. One immunocompromised patient from Mali had RVF detected in urine and semen 74 days post onset of illness [77]. This has been observed with other viral infections such as Ebola and Zika virus disease [78,79].

The RVF virus effectively multiplies in most cells and organs in the body but more so in the liver, brain and spleen [1,8]. Studies have shown that both hepatic and renal injury can occur concurrently [31,32,54] and are a major cause of death. In an RVF rhesus macaque model, the liver exhibited the highest concentration of the virus, implying it is a major site of viral replication [80]. Hepatocyte damage affects the synthesis of clotting factors which may lead to bleeding and prolongation in activated partial thromboplastin time and prothrombin time [80,81]. Haemorrhage is also caused by direct viral invasion and damage to the endothelial cells and thrombocytopenia. The elevation in hepatic transaminases has previously been associated with a fatal outcome [82].

In this review one-fifth (21%) of patients with RVF clinical signs and symptoms succumbed to death and these were mostly those who attended the hospital as in-patients and outpatients. A similar picture of disease severity was reflected across all the other syndromes presented in Table 2. Most of the patients reported in the included studies had severe forms of disease (neurological, hepatic, renal, haemorrhagic and visual syndromes). This is possibly not a representative picture of the disease in the community as the results analysed greatly underestimate the true number of RVF infections. This bias in disease presentation is relevant from a public health stand point considering mild disease is non-fatal and resolves on its own whereas severely sick patients are likely to seek care than those with mild symptoms and hospitalised severe patients more likely to deteriorate to death. The commonest causes of death in this review included isolated acute hepatic or renal failure, compound hepatorenal impairment, and shock within the first week of illness [31,32,40,41]. Much as this mortality is biased towards those with severe disease, and human mortality from RVF has been observed to increase over the years. A systematic review conducted in 2015 by Nanyingi et al reported human mortality during outbreaks ranging from 0.3% to 44.7% [4]. Whether this is related to increasing viral virulence or different biases in case finding and reporting is not known [83], however, it is important to intensify patient care to prevent or avert liver, renal and cardiovascular failure. In this study, we did not analyze results from studies of serological evidence of acute RVF infection so as to provide an estimate of the percentage of clinically relevant cases to the total estimated RVF infections in the population. Included studies reported on the number of acutely ill or symptomatic cases (which would form the numerator) but not on the total number of people that were positive for RVF in the general population (which would form the denominator). Serosurveys during RVF outbreaks would be a valuable adjunct to clinical case focussed studies.

Although pulmonary symptoms were sparsely reported, one study in Egypt reported one possible case of human-human airborne droplets transmission in a medical doctor who examined patients [39]. No other studies have reported on the potential risk of human-human airborne transmission, however, inhalation of infected aerosols is presumed to be a common route for RVF transmission from infected tissues by herders, abattoir, laboratory and veterinary workers [34,83,84]. In mice, aerosol exposure to RVF was associated with an early and severe development of neurological disease than the subcutaneous route [85]. Pulmonary symptoms have been reported to be the most prominent feature of RVF in ferrets [8]. Because of the speed with which airborne transmissions occur, this could be a possible route explored by bioterrorists and thus underlines the need for development and licensure of RVF vaccines and antivirals.

One major complexity found in this review was the concomitance of RVF with other infections such as malaria [34,52], schistosomiasis [39], hepatitis B [22], chikungunya [33], dengue [33], herpes simplex [34] and systemic fungal infections [39]. The independent detrimental effect of these infections to the liver and other body organs, as well as their immunomodulatory

effects are likely to transform the true clinical picture of RVF [86–88]. A robust clinical case definition that covers most of the diverse clinical manifestations of RVF is important as in some instances advanced serology or viral detection tests may be inaccessible [17]. In this review, less than half (14/30) of the studies used a case definition, even then missed 3 to 5 syndromes. This implies many cases of RVF may have been missed in previous outbreaks due to inadequate definition of the disease.

Because of the great diversity in RVF clinical manifestations, we propose a clinical case definition comprising of the general febrile and hepatic syndromes as the cardinal manifestations of RVF disease whether or not associated with the gastrointestinal, renal, neurological, haemorrhagic, obstetric, visual, and cardio-pulmonary syndromes and death. Not all symptoms under each syndrome need to occur together for a conclusion to be drawn on the occurrence of a particular RVF syndrome, but the occurrence of any one of the common symptoms with or without any of the rare symptoms under each syndrome. No single study included in this review captured all the symptoms presented, rather each individual study contributed a portion to the pooled estimate thus the proposed case definition reflects the real context (that is, any one symptom under each syndrome) that originated it. The syndromic approach to RVF diagnosis has several advantages to the bed-side clinicians, epidemiologists investigating outbreaks and public health surveillance. The syndromic categorisation reflects the multisystemic pathogenesis of the disease, is simple, inexpensive and easier to remember and likely to aid in soliciting symptoms that are not spontaneously reported by patients [89]. During an outbreak it can be implemented on a large scale among heath care workers with varied experience and level of training. We believe the inclusion and distinction between common and rare manifestations in the RVF disease case definition provides a detailed repertoire for reference as in some instances the rare manifestations may be the only features of disease in patients. At a public health level, well-structured and long-term syndromic disease surveillance programmes have been found to be effective in the early detection of outbreaks and implementation of care and control measures [90,91]. These programmes depend on reports of non-specific signs and symptoms analysed long before definitive laboratory diagnostic data is available [92].

In RVF endemic countries it would be ideal for human surveillance programmes to embrace the syndromic approach to diagnosis or data generation and analysis so as to inform quick public health responses [17]. Signals for vigilance could be heightened by notifications of increased RVF virus activity from the veterinary and environmental sector surveillance programmes. Sentinel herds monitoring (SHM) of RVF virus activity is a common practice in the animal health sector. Similarly, climatologists use remote sensing satellite data (RSSD) including cold cloud density (CCD), Normalised differentiated vegetation index (NDVI), sea surface temperature (SST) monitoring for the Indian and Pacific Oceans, basin excess rainfall monitoring system (BERMS) measurements and three months rolling mean value (RMV) of rainfall days and quantity for monitoring rainfall patterns. This review showed that human RVF outbreaks have occurred in different countries at different time points in a year, though mostly in March/April and October/November. These results align well with the rainy seasons in East Africa which are known to peak in the same months with occasional intra-seasonal and interannual variability in different countries [93]. This could be significant in planning human RVF surveillance if coupled with information from SHM and RSSD. RMV, SST and BERMS have the capacity to forecast RVF outbreaks 1–2.5 months, 2–5 months and 5months respectively [94]. These epidemiological predictive tools can be used to alert the human ministry to intensify on RVF surveillance ahead of any potential outbreak [83]. Practically this could enhance the currently loose 'One-health' inter-sectoral collaboration [20].

Despite the proposed case definition, it is clinically difficult to differentiate RVF from other endemic tropical infections. In this systematic review and meta-analysis, we included only

laboratory confirmed RVF cases but still no distinct manifestation stands out to be peculiar to RVF. The disease picture overlaps in similarity to that of yellow fever, viral hepatitis, chikungunya, malaria, dengue fever, typhoid, to mention but a few. This implies that the clinical features detailed in this study can only contribute to a clinical suspicion of the disease in humans. It is therefore important to link the clinical suspicion to the epidemiological abortion storms and very high sudden new born deaths in livestock as these seem to be distinct to RVF, although this might not apply among travellers, accidental laboratory infections or in case the pathogen is used as a bioterrorism agent targeting humans. Another challenge is that in some instances RVF outbreaks have occurred in humans without preceding animal loss, meaning these epidemiological links should also be used with caution. Currently, a definitive diagnosis of RVF needs to be made in the laboratory using any of the tests indicated in Table 1. The down side of these tests is that they are done in advanced laboratories located far away from the remote settings where RVF outbreaks occur thus underlines the need for development of rapid diagnostic tests (RDT) [95]. None of the studies included in this review used an RDT as an RVF confirmatory test yet these can detect recent and/or previous exposure in real time of health care worker—patient contact.

The major strength of this review is that it is the first time a pooled estimate of the relative proportion of RVF clinical manifestations and narrative description of clinical syndromes are being concurrently presented. Two previous reviews have reported on the clinical manifestations of RVF in humans [63,96], however, these studies did not incorporate a meta-analysis, which makes it difficult to judge the symptoms that are most common in RVF since the same symptoms are shared by other infections in RVF endemic areas. There were several limitations. Most studies were case series with small sample sizes. This has previously been reported as a major shortfall for RVF studies [97] and likely to hinder phase III clinical trials. Secondly, there was inadequate capture of symptoms in parent studies. Symptoms were either sparsely captured or lumped together for example as haemorrhagic disease, CNS symptoms which probably reflect differences in clinical diagnostic capacity. In this review, we endeavoured to capture the detailed clinical manifestations as some of these may be the only symptoms in patients. Thirdly, we included only studies published in English and it is possible that studies published in other languages could have added more details to this review. Fourth, most studies enrolled mostly in-patients and out-patients which biases the overall picture to a more severe disease. Finally, the effect of treatment prior to capturing their symptoms was not assessed as most febrile patients tend to self-medicate before seeking professional medical care.

In conclusion, RVF disease has a complex symptomatology in humans and we have used the syndromic approach to delineate its presentation. This approach is likely to speed up case detection by health care workers and surveillance teams as well as increase public awareness about RVF. The implementation of syndromic surveillance could enhance the human, animal and environmental 'one-health' inter-sectoral collaborations in disease control.

## Supporting information

**S1 Fig. RVF prevalence among included studies that reported patients by gender.**
[5,8,9,17,21,22,31–37,39–43,45,47–50,53,55].
(PDF)

**S2 Fig. Distribution of RVF outbreaks in a year for included studies.** [5,8,9,17,20–22,31–55].
(PDF)

**S3 Fig. Forest plots for the common symptoms under the general febrile syndrome.** n–number of patients with the sign or symptom; N–total number of patients in the study assessed

for sign or symptom; %—percentage; ES (95% CI)–estimated 95% confidence interval; % weight–percentage weight of the study calculated from random effects meta-analysis; I[2] –chi-square value; p–p-value; Inpatients–subjects source in the study was hospital based patients requiring admission; Outpatients–subjects source in the study was hospital based patients requiring no admission; Inpatients and outpatients -subjects source in the study was both hospital based patients requiring admission and no admission and data collection in the included studies was combined; Community patients-subjects source in the study was non-hospital based patients found in the community or at home. [5,8,9,17,20–22,32–37,39–41,43–46,48,49,50,52,53].
(PDF)

**S4 Fig. Forest plots for the common symptoms under the gastrointestinal syndrome.** n–number of patients with the sign or symptom; N–total number of patients in the study assessed for sign or symptom; %—percentage; ES (95% CI)–estimated 95% confidence interval; % weight–percentage weight of the study calculated from random effects meta-analysis; I[2] –chi-square value; p–p-value; Inpatients–subjects source in the study was hospital based patients requiring admission; Outpatients–subjects source in the study was hospital based patients requiring no admission; Inpatients and outpatients -subjects source in the study was both hospital based patients requiring admission and no admission and data collection in the included studies was combined; Community patients -subjects source in the study was non-hospital based patients found in the community or at home. [5,9,17,20–22,32–37,40,41,43,44,46,48–50,52].
(PDF)

**S5 Fig. Forest plots for the common symptoms under the hepatic syndrome.** n–number of patients with the sign or symptom; N–total number of patients in the study assessed for sign or symptom; %—percentage; ES (95% CI)–estimated 95% confidence interval; % weight–percentage weight of the study calculated from random effects meta-analysis; I[2] –chi-square value; p–p-value; Inpatients–subjects source in the study was hospital based patients requiring admission; Outpatients–subjects source in the study was hospital based patients requiring no admission; Inpatients and outpatients -subjects source in the study was both hospital based patients requiring admission and no admission and data collection in the included studies was combined; Community patients -subjects source in the study was non-hospital based patients found in the community or at home. [5,17,21,22,31,32,34,40,41,44,46,49,50].
(PDF)

**S6 Fig. Forest plots for the common symptoms under the renal syndrome.** n–number of patients with the sign or symptom; N–total number of patients in the study assessed for sign or symptom; %—percentage; ES (95% CI)–estimated 95% confidence interval; % weight–percentage weight of the study calculated from random effects meta-analysis; I[2]– chi-square value; p–p-value; Inpatients–subjects source in the study was hospital bas ed patients requiring admission; Outpatients–subjects source in the study was hospital based patients requiring no admission; Inpatients and outpatients—subjects source in the study was both hospital based patients requiring admission and no admission and data collection in the included studies was combined; Community patients -subjects source in the study was non-hospital based patients found in the community or at home. [5,21,31,32].
(PDF)

**S7 Fig. Forest plots for the common symptoms under the neurological syndrome.** n–number of patients with the sign or symptom; N–total number of patients in the study assessed for sign or symptom; %—percentage; ES (95% CI)–estimated 95% confidence interval; % weight–

percentage weight of the study calculated from random effects meta -analysis; $I^2$ –chi-square value; p–p-value; Inpatients–subjects source in the study was hospital based patients requiring admission; Outpatients–subjects source in the study was hospital based patients requiring no admission; Inpatients and outpatients—subjects source in the study was both hospital based patients requiring admission and no admission and data collection in the included studies was combined; Community patients—subjects source in the study was non-hospital based patients found in the community or at home. [5,8,9,17,22,32,34,39–41,43,44,46,48–50].
(PDF)

**S8 Fig. Forest plots for the common symptoms under the haemorrhagic syndrome.** n–number of patients with the sign or symptom; N–total number of patients in the study assessed for sign or symptom; %—percentage; ES (95% CI)–estimated 95% confidence interval; % weight–percentage weight of the study calculated from random effects meta -analysis; $I^2$ –chi-square value; p–p-value; Inpatients–subjects source in the study was hospital based patients requiring admission; Outpatients–subjects source in the study was hospital based patients requiring no admission; Inpatients and outpatients—subjects source in the study was both hospital based patients requiring admission and no admission and data collection in the included studies was combined; Community patients—subjects source in the study was non-hospital based patients found in the community or at home. [5,8,17,20–22,32,36,37,39,40,41,43,44,48,52,53].
(PDF)

**S9 Fig. Forest plots for the common symptoms under the visual syndrome.** n–number of patients with the sign or symptom; N–total number of patients in the study assessed for sign or symptom; %—percentage; ES (95% CI)–estimated 95% confidence interval; % weight–percentage weight of the study calculated from random effects meta-analysis; $I^2$ –chi-square value; p–p-value; Inpatients–subjects source in the study was hospital based patients requiring admission; Outpatients–subjects source in the study was hospital based patients requiring no admission; Inpatients and outpatients -subjects source in the study was both hospital based patients requiring admission and no admission and data collection in the included studies was combined; Community patients -subjects source in the study was non-hospital based patients found in the community or at home. [8,9,17,21,34,36,37,39–41,43,44,48,49,55].
(PDF)

**S10 Fig. Forest plots for the common symptoms under the cardio-pulmonary syndrome.** n–number of patients with the sign or symptom; N–total number of patients in the study assessed for sign or symptom; %—percentage; ES (95% CI)–estimated 95% confidence interval; % weight–percentage weight of the study calculated from random effects meta-analysis; $I^2$ –chi-square value; p–p-value; Inpatients–subjects source in the study was hospital based patients requiring admission; Outpatients–subjects source in the study was hospital based patients requiring no admission; Inpatients and outpatients—subjects source in the study was both hospital based patients requiring admission and no admission and data collection in the included studies was combined; Community patients—subjects source in the study was non-hospital based patients found in the community or at home. [8,17,44,48].
(PDF)

**S1 Table. Search strategy in Embase database.**
(DOCX)

**S2 Table. Search strategy in Medline database.**
(DOCX)

**S3 Table. Search strategy in Global Health database.**
(DOCX)

**S4 Table. Search strategy in Web of Science database.**
(DOCX)

**S5 Table. Risk of Bias Tool (adopted and modified from the Hoy and Brooks tool).**
(DOCX)

**S6 Table. Risk of bias in included studies for the clinical manifestations of Rift Valley fever (Yes = Low risk; No = High risk).**
(DOCX)

**S7 Table. Rift Valley fever case definitions use and clinical syndromes manifested by patients in the respective outbreaks.**
(DOCX)

**S8 Table. Characteristics of RVF clinical symptoms in humans.**
(DOCX)

**S9 Table. Proportion of patients with other RVF clinical manifestations for which a pooled prevalence could not be estimated.**
(DOCX)

**S1 Appendix. Risk of Bias Tool.**
(PDF)

**S1 PRISMA checklist. PRISMA 2020 Checklist for the clinical manifestations of Rift Valley fever in humans: Systematic review and meta-analysis.**
(DOCX)

## Author Contributions

**Conceptualization:** Zacchaeus Anywaine.

**Data curation:** Zacchaeus Anywaine, Swaib Abubaker Lule.

**Formal analysis:** Zacchaeus Anywaine, Christian Hansen, Alison Elliott.

**Funding acquisition:** George Warimwe.

**Investigation:** Zacchaeus Anywaine, Swaib Abubaker Lule.

**Methodology:** Zacchaeus Anywaine, Christian Hansen, George Warimwe, Alison Elliott.

**Project administration:** Zacchaeus Anywaine, Alison Elliott.

**Resources:** Zacchaeus Anywaine, Alison Elliott.

**Supervision:** Christian Hansen, George Warimwe, Alison Elliott.

**Validation:** Swaib Abubaker Lule, Christian Hansen, George Warimwe, Alison Elliott.

**Visualization:** Swaib Abubaker Lule, Christian Hansen, George Warimwe, Alison Elliott.

**Writing – original draft:** Zacchaeus Anywaine.

**Writing – review & editing:** Zacchaeus Anywaine, Swaib Abubaker Lule, Christian Hansen, George Warimwe, Alison Elliott.

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
