## [Decision Letter · Decision Letter 0]

18 Jul 2021

Dear Dr. Anywaine,

Thank you very much for submitting your manuscript "Clinical manifestations of Rift Valley fever in humans: Systematic review and meta-analysis" for consideration at PLOS Neglected Tropical Diseases. As with all papers reviewed by the journal, your manuscript was reviewed by members of the editorial board and by several independent reviewers. In light of the reviews (below this email), we would like to invite the resubmission of a significantly-revised version that takes into account the reviewers' comments. 

We cannot make any decision about publication until we have seen the revised manuscript and your response to the reviewers' comments. Your revised manuscript is also likely to be sent to reviewers for further evaluation.

Sincerely,

Anita K. McElroy, MD, PhD

Associate Editor

A. Desiree LaBeaud

Deputy Editor

Reviewer's Responses to Questions

**Key Review Criteria Required for Acceptance?**

**Methods**

-Are the objectives of the study clearly articulated with a clear testable hypothesis stated?

-Is the study design appropriate to address the stated objectives?

-Is the population clearly described and appropriate for the hypothesis being tested?

-Is the sample size sufficient to ensure adequate power to address the hypothesis being tested?

-Were correct statistical analysis used to support conclusions?

-Are there concerns about ethical or regulatory requirements being met?

Reviewer #1: Methods and statistical analyses are appropriate to the scope of the study.

Reviewer #2: • Table 2 requires clarification of using the term “In- and Out-patients” separately from the “In-patients” or “Out-patients”. Please explain the rationale in text.

Reviewer #3: The objectives of the study were clearly stated in the author summary and introduction of the manuscript. The authors state that the study is to conduct a “systematic review and meta-analysis of existing literature

aimed at determining the frequency and scope of clinical and laboratory manifestations of RVF in humans”.

In addition the authors highlighted and outlined 5 main learning points for their analysis. 

The study design was a systematic review of literature and was done using standardized protocols and was appropriate given the stated objectives. The methodology and revie3w of manuscripts to include in the systematic review was also appropriate and in total 32 manuscripts were used for data extraction and analysis. Although this seems to be a small number it is likely there is not as an extensive body of literature containing the specific requirements laid out by the authors for data to include. And as the authors state in the discussion, there are likely more non-English papers containing relevant data but were not included in this analysis. 

No ethical concerns were noted.

**Results**

-Does the analysis presented match the analysis plan?

-Are the results clearly and completely presented?

-Are the figures (Tables, Images) of sufficient quality for clarity?

Reviewer #1: Results are presented clearly and adequately.

Reviewer #2: • Figure legends are not found for all images.

Reviewer #3: The analysis of the systematic review and clinical case data was appropriate and straight forward. All analysis and results were descriptive of the findings from the manuscripts reviewed. As the authors stated, most for the clinical manifestations described would likely be from patients with more severe or late stage RVF disease. Although this may bias the presentation of results it does seem relevant from a public health standpoint considering mild illness is non-fatal and resolves on its own, but the results analyzed will greatly underestimate the true number of RVF infections. The authors did not analyze or compare results or studies of serological evidence of RVF infection as a comparison to the acutely ill case studies to provide an estimate of what percent of the clinically relevant cases presented in this manuscript to the total estimated RVF infections in the population. 

The tables and figures accurately represent the data presented and are thorough and detailed.

**Conclusions**

-Are the conclusions supported by the data presented?

-Are the limitations of analysis clearly described?

-Do the authors discuss how these data can be helpful to advance our understanding of the topic under study?

-Is public health relevance addressed?

Reviewer #1: Conclusions summarized in “Key learning points” require attention as indicated in the Summary and General Comments of the review.

Reviewer #2: Clearly described the significance of this metaanalysis.

Reviewer #3: The overall discussion and conclusions stated by the authors agrees with the data presented. In general the authors discuss the complexity and broad range of clinical manifestations and signs and symptoms associated with RVF infection. The authors also explain that the RVF clinical picture can be associated with many other more common tropical infections and may not always be recognized at first as suspect RVF when cases initially seek medical care. 

The authors do recognize this analysis does have some limitations, They state that most of the manuscripts included in the analysis would be considered the more severe end of the RVF clinical spectrum, thus biasing the analysis and clinical data associated with RVF infection. They also state that most studies used in the analysis are small sample size and limited in detailed clinical data. In addition, the authors mention that this was not a meta analysis and limiting identification of symptoms more common to RVF. 

The authors spend much of the discussion detailing the biological, immunological or virological explanations for the clinical manifestations identified in their analysis. Although this does add some good context and background for these clinical manifestations, I did not think the authors focused enough on the real-world relevance of their findings. The authors mention that this analysis would help improve routine surveillance, case finding and “one-health” policies. There was no detailed discussion related to these points. In fact the authors state that proposing a case definition form the analysis would be “reductionistic”. I think the authors could have expanded on the public health and surveillance relevance of their findings to how to improve and identify the most common clinical manifestations of RVF and helped propose which clinical manifestations would be best to prioritize suspect RVF cases. 

The authors also listed 5 “Key Learning Points” at the beginning of their manuscript. 

1. Human RVF outbreaks can occur any time of the year in endemic areas and over the years there has been an increased reporting of the disease among women.

2. RVF presents with diverse clinical manifestation which are inherently related to the nature of the infecting virus strains hence human disease is best defined inform of clinical syndromes.

3. A clear and detailed definition of human disease is likely to improve case detection during routine surveillance and outbreaks, estimation of disease burden, and consequently inform “one-health” policies in endemic countries.

4. Gaps exist on the link between RVF and human obstetric outcomes, the ability to induce autoimmunity, and its persistence in body tissues beyond what is currently known.

5. Airborne transmission is possible thus underlines the need for development and licensure of RVF vaccines and antivirals.

The authors do address all of these points in their paper, but I would question inclusion of some based on the low priority in which they were discussed in the manuscript. The authors provided detailed explanation of points #2 and #4 and to a lesser extent #1. I feel points # 3 and #5 were not sufficiently supported. A stated above, the authors did not propose or discuss a detailed case definition or how one based on their findings would improve surveillance or one health policies. They in fact stated the opposite and suggested the clinical manifestations were too complex and broad to do so and a syndromic approach would be best. For point #5, the only mention of this in the discussion was a single case from Egypt. This could be argued, if in fact true, is a very rare occurrence and could have been caused by a number of clinical procedures causing aerosolized RVF to infect a physician. This does not seem to be a primary learning point and not a major finding of this analysis.

**Editorial and Data Presentation Modifications?**

Reviewer #1: No editorial suggestions.

Reviewer #2: • Line 91: “In the year 2000, a major outbreak occurred for the first time in the Arabian Peninsula [5, 6], and human cases have been reported in non-endemic countries such as the United States of America (USA) [7, 8], United Kingdom (UK) [9], and China [10]”. This sentence should be separated into two, because those listed RVF cases in the U.S. reported in 1930s.

• Line 329: “The NSs protein also forms filamentous structures that interact with nuclear chromatin to cause chromosomal segregation and defects [59, 60] and this is believed to be the mechanism for the congenital anomalies and abortion storms in animals” Although the NSs proteins are considered virulence factor for RVF, the direct association with the abortion has not been demonstrated. Rather, Oymans J. et al. (PLoS NTD, 2020, 14: e0007898) demonstrated that RVFV infection in trophoblasts can trigger inflammation and hemorrhage in the placenta in sheep, which is likely the cause of abortion.

• Line 362: “In an RVF rhesus macaque model, the kidneys exhibited the highest concentration of the virus, implying it is a major site of viral replication [76].” This study showed RVFV titers in kidneys lower than those in livers or spleens, in which authors stated that “Titration of organ homogenates clearly established liver and spleen as major sites of viral replication, with kidney and lung also suspect”. Accordingly, authors should correct the interpretation.

• The RVF patient returned to China was initially misdiagnosed as Yellow Fever. It might be important to indicate how to distinguish RVF from other similar febrile illnesses.

Reviewer #3: none

**Summary and General Comments**

Reviewer #1: The review by Anywaine et al. on clinical manifestations of RVF in humans based on systematic review and meta-analysis is a useful contribution that outlines the complex symptomatology of human RVF disease into syndromes. This approach might assist clinicians and epidemiologist in defining the disease case definition, thus has a potential to improve RVF detection rates, syndromic disease surveillance, outbreaks control, and increase public awareness. 

The authors should consider not overstating frequency of severe symptoms in RVFV-infected humans. Description of RVF as a “mosquito-borne viral haemorrhagic zoonosis associated with severe morbidity” undermines the common clinical course of RVFV infection in humans, which is mostly moderate self-limiting febrile disease.

More specific comments to be addressed:

“Human symptoms have poor specificity” it should be rather “Clinical manifestations (signs and symptoms) in human are unspecific”

Authors claim that they “determined the clinical manifestations of RVF in humans” where in fact they review and analyzed them.

“Death occurred in 21% (95% CI 14-29; [16 studies, 328 patients]) of cases. These results should be more carefully or more responsibly interpreted because they imply relatively very high case fatality rate among RVF-infected people. It should explained that these results mostly concern hospitalized cases. Most community infections with RVFV are asymptomatic or mild. 

In “Author summary: the authors stated that “limited surveillance and estimation of disease burden ….is due to the inability to concisely define the disease” This is too simplified. One of the major problem in RVF-endemic countries with diagnosis of RVF (notably Africa) is limited diagnostic capacity, lack of well-structured and long-term surveillance studies, including inter-epidemic periods and long inter-epidemic periods that make justification for such as studies difficult. 

The “Key learning points” require most attention. 

“Human RVF outbreaks can occur any time of the year in endemic areas” 

Is this true? Are RVFV vectors active all the year around in all RVF-endemic regions?

“RVF presents with diverse clinical manifestation which are inherently related to the nature of the

infecting virus strains” 

This statement suggest that diversity of clinical manifestations depend on specific nature/ability of a RVFV strain involved or that different RVFV strains are responsible for different clinical outcomes in humans. Is this supported by any study in humans?

“A clear and detailed definition of human disease is likely to improve case detection during routine

surveillance and outbreaks, estimation of disease burden, and consequently inform “one-health” policies” 

The later conclusion on how it will inform “One Health” policies is unclear and should be explain. 

“Gaps exist on the link between RVF and human obstetric outcomes, the ability to induce autoimmunity,

and its persistence in body tissues beyond what is currently known”. 

The review is not supporting or is not giving sufficient background/facts to justify that there are knowledge gaps and thus a need for studies on RVFV ability to induce autoimmunity disease or to establish a persistence /chronic state in body tissues.

“Airborne transmission is possible thus underlines the need for development and licensure of RVF

vaccines and antivirals” 

Again, authors overemphasize the RVF as a formidable public health threat. This is not necessary to justify its public health importance. Also, note that RVF epidemic-prone potential is not driven by airborne transmission, thus “airborne transmission” is misused to justify the need for the development of vaccines and antivirals.

Reviewer #2: The manuscript entitled “Clinical manifestations of Rift Valley fever in humans: Systematic review and metaanalysis” by Anywaine Z., et al. analyzed selected 32 reports for the meta-analysis of common RVF clinical signs. This study was performed to improve the poor RVF case definition used to analyze past RVF outbreaks. Overall, this manuscript is well written and captured representative RVF symptoms with the estimated proportions. There are several points requiring further clarifications.

Reviewer #3: Overall, this manuscript is a good review of available English language manuscripts on RVF clinical manifestations of illness. It does consolidate a broad range of findings and summarizes them well. I suggest the discussion expand on the real-work and public health relevance of the findings more and revise the key learning points.

PLOS authors have the option to publish the peer review history of their article (what does this mean?). If published, this will include your full peer review and any attached files.

Reviewer #1: No

Reviewer #2: No

Reviewer #3: No
---

## [Decision Letter · Decision Letter 1]

4 Jan 2022

Dear Dr. Anywaine,

Thank you very much for submitting your manuscript "Clinical manifestations of Rift Valley fever in humans: Systematic review and meta-analysis" for consideration at PLOS Neglected Tropical Diseases. As with all papers reviewed by the journal, your manuscript was reviewed by members of the editorial board and by several independent reviewers. The reviewers appreciated the attention to an important topic. Based on the reviews, we are likely to accept this manuscript for publication, providing that you modify the manuscript according to the review recommendations. 

Sincerely,

Anita K. McElroy, MD, PhD

Associate Editor

A. Desiree LaBeaud

Deputy Editor

Reviewer's Responses to Questions

**Key Review Criteria Required for Acceptance?**

**Methods**

-Are the objectives of the study clearly articulated with a clear testable hypothesis stated?

-Is the study design appropriate to address the stated objectives?

-Is the population clearly described and appropriate for the hypothesis being tested?

-Is the sample size sufficient to ensure adequate power to address the hypothesis being tested?

-Were correct statistical analysis used to support conclusions?

-Are there concerns about ethical or regulatory requirements being met?

Reviewer #2: Authors revised unclear points in Methods section according to reviewers' suggestions.

Reviewer #4: The objectives of the study were clearly stated.

**Results**

-Does the analysis presented match the analysis plan?

-Are the results clearly and completely presented?

-Are the figures (Tables, Images) of sufficient quality for clarity?

Reviewer #2: Authors added Figure legends appropriately.

Reviewer #4: Results were clear - the figures were not of sufficient quality to review.

**Conclusions**

-Are the conclusions supported by the data presented?

-Are the limitations of analysis clearly described?

-Do the authors discuss how these data can be helpful to advance our understanding of the topic under study?

-Is public health relevance addressed?

Reviewer #2: The manuscript has been improved for clarity after the revision.

Reviewer #4: Conclusion was adequate.

**Editorial and Data Presentation Modifications?**

Reviewer #2: Adequately addressed all points.

Reviewer #4: Minor Revision.

**Summary and General Comments**

Reviewer #2: The revised manuscript by Anywaine Z. et al. was adequately addressed this reviewer's points.

Reviewer #4: Please see uploaded document.

PLOS authors have the option to publish the peer review history of their article (what does this mean?). If published, this will include your full peer review and any attached files.

Reviewer #2: No

Reviewer #4: Yes: Orienka Hellferscee

Figure Files:

Data Requirements:

Reproducibility:

References

---

## [Editor Report · Decision Letter 2]

3 Feb 2022

Dear Dr. Anywaine,

We are pleased to inform you that your manuscript 'Clinical manifestations of Rift Valley fever in humans: Systematic review and meta-analysis' has been provisionally accepted for publication in PLOS Neglected Tropical Diseases.

Best regards,

Anita K. McElroy, MD, PhD

Associate Editor

A. Desiree LaBeaud

Deputy Editor

---

## [Editor Report · Acceptance letter]

17 Mar 2022

Dear Dr. Anywaine,

We are delighted to inform you that your manuscript, "Clinical manifestations of Rift Valley fever in humans: Systematic review and meta-analysis," has been formally accepted for publication in PLOS Neglected Tropical Diseases.

Best regards,

Shaden Kamhawi

co-Editor-in-Chief

Paul Brindley

co-Editor-in-Chief
